# WHY IS SAM ROBUST TO LABEL NOISE?

**Christina Baek**   **Zico Kolter**   **Aditi Raghunathan**
Carnegie Mellon University
{kbaek, zkolter, raditi}@andrew.cmu.edu

## ABSTRACT

Sharpness-Aware Minimization (SAM) is most known for achieving state-of the-art performances on natural image and language tasks. However, its most pronounced improvements (of tens of percent) is rather in the presence of label noise. Understanding SAM's label noise robustness requires a departure from characterizing the robustness of minimas lying in "flatter" regions of the loss landscape. In particular, the peak performance under label noise occurs with early stopping, far before the loss converges. We decompose SAM's robustness into two effects: one induced by changes to the logit term and the other induced by changes to the network Jacobian. The first can be observed in linear logistic regression where SAM provably up-weights the gradient contribution from clean examples. Although this explicit up-weighting is also observable in neural networks, when we intervene and modify SAM to remove this effect, surprisingly, we see no visible degradation in performance. We infer that SAM's effect in deeper networks is instead explained entirely by the effect SAM has on the network Jacobian. We theoretically derive the implicit regularization induced by this Jacobian effect in two layer linear networks. Motivated by our analysis, we see that cheaper alternatives to SAM that explicitly induce these regularization effects largely recover the benefits in deep networks trained on real-world datasets.

## 1 INTRODUCTION

In recent years, there has been growing excitement about improving the generalization of deep networks by regularizing the sharpness of the loss landscape. Among optimizers that explicitly minimize sharpness, Sharpness Aware Minimization (SAM) (Foret et al., 2020) garnered popularity for achieving state-of-the-art performance on various natural image and language benchmarks. Compared to stochastic gradient descent (SGD), SAM provides consistent improvements of several percentage points. Interestingly, a less widely known finding from Foret et al. (2020) is that SAM's most prominent gains lie elsewhere, in the presence of random label noise. In fact, SAM is more robust to label noise than SGD by tens of percentage points, rivaling the current best label noise techniques (Jiang et al., 2017; Zhang et al., 2017; Arazo et al., 2019). In Figure 1, we demonstrate this finding in CIFAR10 with 30% label noise, where SAM's best test accuracy is 17% higher. In particular, we find that the robustness gains are most prominent in a particular version of SAM called 1-SAM which applies the perturbation step to each sample in the minibatch separately.

An important caveat about the random label noise setting is that the test accuracy does not improve with further training, but instead peaks in the middle. Consequently, we argue understanding SAM in this regime requires a departure from reducing SAM to the sharpness of its solution at convergence, but instead reasoning about SAM's "early learning" behavior. In fact, even in settings with a unique minimum, the best test accuracy may change based on the optimization trajectory. Indeed, the performance achieved by SAM does not diminish with underparametrization, with the gains above SGD sometimes increasing with more data (Appendix F).

In this work, we investigate why 1-SAM is more robust to label noise than SGD at a more mechanistic level. Decomposing the gradient of each example ("sample-wise" gradient) by chain rule into $\nabla_w \ell(f(w, x), y) = \partial \ell / \partial f \cdot \nabla_w f$, we analyze the effect of SAM's perturbation on the terms $\partial \ell / \partial f$ ("logit scale") and $\nabla_w f$ ("network Jacobian"), separately. We make the following key conclusions about how these components improve early-stopping test accuracy. We begin our study in linear models, where the only difference between SAM and SGD is the logit scale term. Here, we show that SAM reduces to a reweighting scheme that *explicitly* up-weights the gradient contribution of

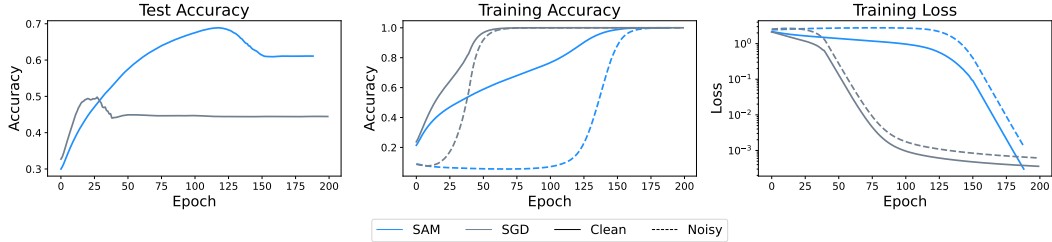

Figure 1: CIFAR10 training accuracy and loss in clean versus noisy data. SAM achives a higher clean training accuracy before fitting the noisy data, i.e., when accuracy of noisy training data surpasses random chance. This corresponds with a higher peak in test accuracy.

low loss points. This effect is particularly useful in the presence of mislabeled examples. When training with gradient descent, correctly labeled or *clean* examples initially dominate the direction of the update and as a result, their corresponding loss decreases first before that of mislabeled or *noisy* examples (Liu et al., 2020; 2023). Similar to many existing label-noise techniques (Liu et al., 2020), SAM's explicit up-weighting keeps the gradient contribution of clean examples large even after they are fit, slowing down the rate at which noisy examples are learned in comparison. We hypothesize that higher peak test accuracy corresponds with achieving higher clean training before overfitting to noise. Empirically, the gap between the training accuracy of clean and noisy examples closely tracks the test accuracy in earlier epochs (Figure 2).

In deep networks, SAM's logit scale term empirically induces a similar effect of explicitly up-weighting the gradients of low loss examples. However, for the magnitude of SAM's perturbation utilized in practice, we find that SAM's logit scale has negligible impact on SAM's robustness. On the other hand, just keeping SAM's perturbation on the network Jacobian term *retains nearly identical performance* to SAM. This suggests that there is a fundamentally different mechanism that originates in SAM's Jacobian term that results in most of SAM's label noise robustness in the nonlinear case.

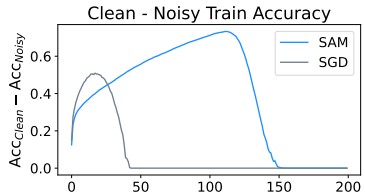

Figure 2: SAM learns clean examples faster than noisy examples.

Motivated by this finding, we analyze the Jacobian-only SAM for a 2-layer linear network, and show that the resulting update decomposes into SGD with $\ell_2$ regularization on the final layer weights and intermediate activations. We argue that the benefits of such regularization could partially be reasoned about in terms of gradient contribution or as an *implicit* up-weighting scheme. Namely, it keeps the loss of correctly fit points high by constraining the magnitude of the network output. Furthermore, including just this regularization for deep networks, while not achieving the full benefits of SAM, nonetheless substantially improves the performance of SGD. We emphasize that these methods are meant to be illustrative, and not intended to be a standalone method for label noise robustness. But in this vein, our findings suggest that SAM's label noise robustness may not come via sharpness properties at convergence, but rather from the optimization trajectory taken.

## 2 PRELIMINARIES

### 2.1 PROBLEM SETUP

**Model**   We consider binary classification. The sign of the model's output $f : \mathbb{R}^d \to \mathbb{R}$ maps inputs $x \in \mathcal{X}$ to discrete labels $t \in \{-1, 1\}$. We will study two kinds of models – a linear model and a 2-layer deep linear network (DLN) in this work. We do not include the bias term.

$$\textbf{Linear: } f(w, x) = \langle w, x \rangle$$
$$\textbf{DLN: } f(v, W, x) = \langle v, Wx \rangle \text{ where } W \in \mathbb{R}^{d \times h}, v \in \mathbb{R}^h, \tag{2.1}$$

Let us denote the intermediate activation as $z = Wx \in \mathbb{R}^h$. We will abuse the notation to also refer to a generic parameterized model as $f(w, x)$ when clear from context.

**Objective** We consider the logistic loss $\ell(w, x, t) = -\log(\sigma(t \cdot f(w, x)))$ for sigmoid function $\sigma(z) = \frac{1}{1+\exp(-z)}$. Given $n$ training points $[(x_i, t_i)]_{i=1}^n$ sampled from the data distribution $\mathcal{D}$, our training objective is

$$\min_w L(w) \text{ where } L(w) = \frac{1}{n} \sum_{i=1}^n \ell(w, x_i, t_i) \tag{2.2}$$

By chain rule, we can write the sample-wise gradient with respect to the logistic loss $\ell(w, x, t)$ as

$$-\nabla_w \ell(x, t) = t \cdot \underbrace{\sigma(-tf(w, x))}_{\text{logit scale}} \underbrace{\nabla_w f(w, x)}_{\text{Jacobian}} \tag{2.3}$$

The logit scale, which is the model's confidence that $x$ belongs in the incorrect class $-t$, *scales* the Jacobian term. The logit scale grows monotonically with the loss. We refer to the network Jacobian $\nabla_w f(w, x)$ as the "Jacobian term".

Although our mathematical analysis is for binary classification, the cross-entropy loss for multiclass classification observes a similar decomposition for its sample-wise gradient:

$$-\nabla_w \ell(x, y) = \underbrace{\langle e_t - \sigma(f(w + \epsilon_i, x)),}_{\text{logit scale}} \underbrace{\nabla_w f(w, x) \rangle}_{\text{Jacobian}} \tag{2.4}$$

where $\sigma(\cdot)$ is the softmax function and $e_t$ is the one-hot encoding of the target label. Empirically, we will observe that the conclusions from our binary analysis transfer to multi-class.

## 2.2 SHARPNESS AWARE MINIMIZATION

Sharpness-aware Minimization (SAM) (Foret et al., 2020) attempts to find a flat minimum of the training objective (Eq. 2.2) by minimizing the following objective

$$\min_w \max_{\|\epsilon\|_2 \le \rho} L(w + \epsilon), \tag{2.5}$$

where $\rho$ is the magnitude of the adversarial weight perturbation $\epsilon$. The objective tries to find a solution that lies in a region where the loss does not fluctuate dramatically with any $\epsilon$-perturbation. SAM uses first-order Taylor approximation of the loss to approximate worst-case $\epsilon$ with the normalized gradient $\rho \nabla_w L(w)/\|\nabla_w L(w)\|$.

**1-SAM** However, the naive SAM update that computes a single $\epsilon$ does not observe performance gains over SGD in practice unless paired with a small batch size (Foret et al., 2020; Andriushchenko & Flammarion, 2022a). Alternatively Foret et al. (2020) propose sharding the minibatch and calculating SAM's adversarial perturbation $\epsilon$ for each shard separately. At the end of this extreme is 1-SAM which computes $\epsilon$ for the loss of each example in the minibatch individually. Formally, this can be written as

$$w = w - \eta \left( \frac{1}{n} \sum_{i=1}^n \nabla_{w+\epsilon_i} \ell(w + \epsilon_i, x_i, t_i) \right) \text{ where } \epsilon_i = \rho \frac{\nabla_w \ell(x_i, t_i)}{\|\nabla_w \ell(x_i, t_i)\|_2} \tag{2.6}$$

In practice, 1-SAM is the version of SAM that achieves the most performance gain. In this paper, we focus on understanding 1-SAM and will use SAM to refer to 1-SAM unless explicitly stated otherwise.

## 2.3 HYBRID 1-SAM

In our study of 1-SAM, we try to isolate the robustness effects of 1-SAM coming from how the perturbation $\epsilon_i$ affects the logit scaling term and the Jacobian term of each sample-wise update. To do so, we will also pay attention to the following variants of 1-SAM:

$$\text{1-SAM}: \nabla_{w+\epsilon_i} \ell(w + \epsilon_i, x_i, t_i) = t\sigma(-tf(\boxed{w+\epsilon_i}, x))\nabla_{w+\epsilon_i} f(\boxed{w+\epsilon_i}, x)) \tag{2.7}$$

$$\text{Logit SAM}: \Delta^{\text{L-SAM}} \ell(w, x_i, t_i) = t\sigma(-tf(\boxed{w+\epsilon_i}, x))\nabla_w f(\boxed{w}, x_i) \tag{2.8}$$

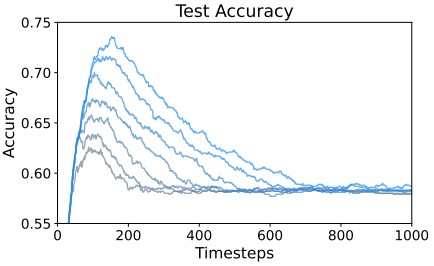 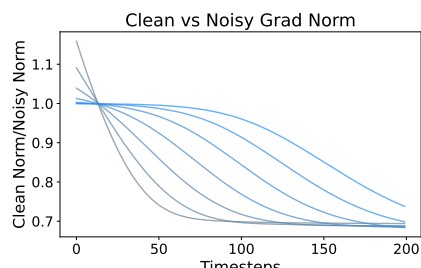

(a) SAM test accuracy for $\rho \in [0, 0.18]$. Bluer curves denote larger $\rho$. Accuracy improves with larger $\rho$.

(b) Ratio of the average sample-wise gradient norm of clean versus noisy points

Figure 3: Linear models trained on the toy Gaussian data using SAM. SAM's preferential up-weighting of low loss points (right) corresponds with higher early stopping test accuracy (left).

$$\text{Jacobian SAM}: \Delta^{\text{J-SAM}}\ell(w, x_i, t_i) = t\sigma(-tf(\boxed{w}, x_i))\nabla_{w+\epsilon_i} f(\boxed{w + \epsilon_i}, x_i) \tag{2.9}$$

Logit SAM (L-SAM) only applies the SAM perturbation to the logit scale for each sample-wise gradient while Jacobian SAM (J-SAM) only applies the SAM perturbation to the Jacobian term. We will observe that in deep networks, J-SAM observes close to the same performance as 1-SAM while L-SAM does not.

## 3 LINEAR MODELS: SAM UP-WEIGHTS THE GRADIENTS OF LOW LOSS EXAMPLES

We first study the robustness affects of the logit scale term in linear models. Not only can SAM's boost in early-stopping performance be observed in linear models, but the Jacobian term $\nabla_w f(w, x) = x$ for linear models and is independent of the weights. Thus, any robustness obtained by SAM in this setting is specifically due to SAM's logit scale.

### 3.1 EXPERIMENTAL INVESTIGATION INTO SAM EFFECT IN LINEAR MODELS

We train our linear models over the following toy Gaussian data distribution $\mathcal{D} : \mathbb{R}^d \times \{-1, 1\}$.

$$\text{True Label: } y \sim \{-1, 1\} \text{ by flipping fair coin}$$
$$\text{Input: } x \sim y \cdot \left[ B \in \mathbb{R}, z \sim \mathcal{N}\left(0, \frac{\gamma^2}{d-1}I_{d-1 \times d-1}\right) \right] \tag{3.1}$$
$$\text{Target: } t = y \cdot \varepsilon \text{ where } \varepsilon \sim \{-1 \text{ w.p } \Delta, \ 1 \text{ w.p } (1-\Delta)\}$$

In $x$, the first dimension contains the true signal $yB$ while the remaining $d - 1$ dimensions is uncorrelated Gaussian noise. We sample training datapoints $(x, t)$ from this distribution where the random label noise $\varepsilon$ corrupts $\Delta$ of the training data. We expect $\Delta < 0.5$ meaning the *majority* of the data is still clean. In the test data, we assume there are no mislabeled examples ($t = y$).

In Figure 3, we compare the performance of full-batch gradient descent and SAM on our toy Gaussian data with $40\%$ label noise (See Appendix A for experimental details). Even in this simple setting SAM observes noticeably higher early stopping test accuracy over SGD. We run a grid search for $\rho$ between 0 and 0.18 and observe that the early stopping test accuracy monotonically increases with $\rho$. Correlated with this improved performance, when we plot the average sample-wise gradient norm of clean examples versus noisy examples along training, we observe that this ratio decays slower with larger $\rho$. This leads us to suspect that SAM's superior test performance is due to clean examples dominating the gradient update for longer. Previously, Liu et al. (2020) proved that at the beginning of gradient descent training in linear models, the clean majority dominates the gradient, causing the clean examples to be fit first. However, the dynamic quickly changes as training progresses. As the loss of clean points quickly decays, the contribution of noisy outliers to the gradient update begins to outweigh that of the clean examples. This results in the test accuracy dropping back down.

In the next section, we will prove that SAM, by virtue of its adversarial weight perturbation, *preferentially* up-weights the gradient signal from low-loss points. This keeps the gradient contribution from clean points high in the early training epochs. Consequently, SAM achieves higher test accuracy by prioritizing fitting more clean training data before overfitting to noise.

## 3.2 ANALYSIS OF SAM'S LOGIT SCALE

Recall that 1-SAM update of datapoint $(x_i, t_i)$ is the gradient evaluated at the weights perturbed by the normalized sample-wise gradient. In linear models, the perturbation is simply the example scaled by the label $\epsilon_i = -\rho t_i \frac{x_i}{\|x_i\|}$. Evaluating the gradient at this perturbed weight, SAM's update reduces to a *constant adjustment* of the logit scale by the norm of the datapoint.

$$-\nabla_{w+\epsilon_i} \ell(w + \epsilon_i, x_i, y_i) = t_i \sigma(-t_i < w, x_i > + \underbrace{\rho\|x_i\|_2}_{\text{constant adjustment}})x_i \tag{3.2}$$

Since the sigmoid function $\sigma$ increases monotonically, the extra positive constant increases the gradient norm of *all* training points. However, among points of the same norm $\|x_i\|$, this adjustment causes low loss points where the confidence towards the incorrect class $\sigma(-t_i\langle w, x_i\rangle)$ is small to be up-weighted *by a larger coefficient* than high loss points where the incorrect class confidence is high, as proven in the following lemma.

**Lemma 3.1 (Preferential up-weighting of low loss points)** *Consider the following function.*

$$f(z) = \frac{\sigma(-z + C)}{\sigma(-z)} = \frac{1 + \exp(z)}{1 + \exp(z - C)} \tag{3.3}$$

*This function is strictly increasing if $C > 0$.*

**Proof** The first derivative is non-negative.

$$\frac{df}{dz} = \frac{\exp(z)}{1 + \exp(z - C)} - \frac{(1 + \exp(z))\exp(z - C)}{(1 + \exp(z - C))^2} \tag{3.4}$$

$$= \frac{\exp(z)(1 - \exp(-C)}{(1 + \exp(z - C))^2} > 0 \quad \forall z \in \mathbb{R} \tag{3.5}$$

∎

We can interpret 1-SAM as a gradient reweighting scheme, where the weight for example $x_i$ is set to

$$\frac{\|\nabla_{w+\epsilon_i} \ell(w + \epsilon_i, x_i, t_i)\|}{\|\nabla_w \ell(w, x_i, t_i)\|} = \frac{\sigma(-t_i\langle w, x_i\rangle + \rho\|x_i\|)}{\sigma(-t_i\langle w, x_i\rangle)}. \tag{3.6}$$

We directly apply Lemma 3.1 by setting $z = t_i\langle w, x_i\rangle$ and $C = \rho\|x_i\|_2$ to prove that across points of the same norm, low loss points are more affected by the up-weighting than high loss points. Since the loss decreases faster for clean points over noisy points, SAM preferentially up-weights the gradients of clean points.

The fact that the resulting early stopping test accuracy of SAM is higher than gradient descent follows by induction. For our toy data distribution, we note that test accuracy saturates as $\rho$ goes to infinity. In particular, 1-SAM converges to the Bayes optimal solution. Furthermore, consistent with previous literature, we find that even in this linear setting, 1-SAM behaves very differently from the naive SAM update (Section 2.2); the latter does not achieve any performance gains from SGD. We further discuss the asymptotic behavior of 1-SAM and the behavior of n-SAM in Appendix B.

## 4 NEURAL NETWORKS: SAM'S ROBUSTNESS COMES FROM THE JACOBIAN

From our linear analysis, we learned that SAM's logit scale term preferentially up-weights the gradients of low-loss points. We verify that this effect of SAM is also visible in neural networks. While the effect persists, we will show that, on the contrary, 1-SAM's logit scale component is not the main reason for SAM's improved robustness in deep networks. In particular, just applying the 1-SAM perturbation to the Jacobian term (J-SAM) recovers most of 1-SAM's gains while logit reweighting alone (L-SAM) cannot. Finally, we find that a large proportion of the performance gains can be recovered through a simpler regularization that mimics the effect of the 1-SAM Jacobian on just the last layer weights and activations.

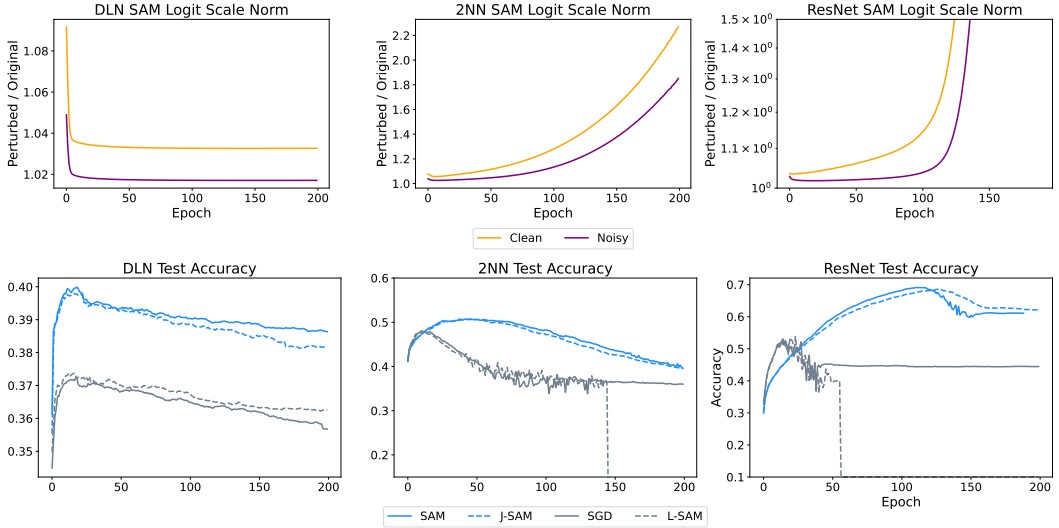

Figure 4: In 2-layer deep linear networks (DLN), 2-layer MLP with ReLU activation (2NN), and ResNet18 trained on noisy CIFAR10, we observe that SAM's perturbation to the logit scale preferentially upweights the gradient norm for clean examples (top row). Yet, J-SAM i.e. SAM absent the explicit reweighting effect, preserves SAM's label noise robustness (bottom row).

## 4.1 EXPLICIT REWEIGHTING DOES NOT FULLY EXPLAIN SAM'S GAINS

We first note that the upweighting of low-loss points can be observed even in multi-class classification with neural networks. Recall the general form of the gradient for multi-class cross-entropy loss is $\nabla_w \ell(x, t) = \langle \sigma(f(w, x)) - e_t, \nabla_w f(w, x) \rangle$ where $\sigma$ is the softmax function and $e_t$ is the one-hot encoding of the label $t$. The analogous "logit scale" component for multiclass classification is the $K$ dimensional vector $g(w, x, t) = \sigma(f(w, x)) - e_t$ whose $\ell_2$ norm scales with the loss. In Figure 4, we measure the change in the norm of this quantity for each example $x_i$ in the minibatch after the SAM perturbation, i.e., $\|g(w + \epsilon_i, x_i, t_i)\| / \|g(w, x_i, t_i)\|$. In ResNet18 trained on CIFAR10 using SAM, we indeed observe that this quantity is higher in clean examples than noisy examples (Figure 4).

This result alone seemingly implies that the explicit reweighting also explains SAM's robustness in deeper models. However, when we isolate this effect by applying SAM to just the logit scale term, i.e., L-SAM, with the optimal perturbation magnitude found for SAM $\rho = 0.01$, we observe marginal early-stopping performance gains above SGD (Figure 4). Alternatively, J-SAM recovers almost all of the gains of SAM. This suggests that SAM's reweighting of the logit scale is not the main contributor to SAM's robustness *in neural networks*. We also find that this observation does not require an arbitrarily deep network and also holds true in simple 2-layer deep linear networks (DLN) and ReLU MLP's (2NN) trained on flattened CIFAR10 images with 30% label noise. A similar analysis of SAM under label noise in linear models was conducted by Andriushchenko & Flammarion (2022b), however they attribute the label noise robustness in neural networks to logit scaling. We claim the opposite: that the *direction* or the network Jacobian of SAM's update becomes much more important.

## 4.2 ANALYSIS

Motivated by this, we study the effect of perturbing the Jacobian in a simple 2-layer DLN. In the linear case, the Jacobian term was constant, but in the nonlinear case the Jacobian is also sensitive to perturbation. In particular, for 2-layer DLNs, J-SAM regularizes the norm of the intermediate activations and last layer weights, as proven by the following proposition.

**Proposition 4.1** *For binary classification in a 2-layer deep linear network $f(v, W, x) = \langle v, Wx \rangle$, J-SAM approximately reduces to SGD with L2 norm penalty on the intermediate activations and last layer weights.*

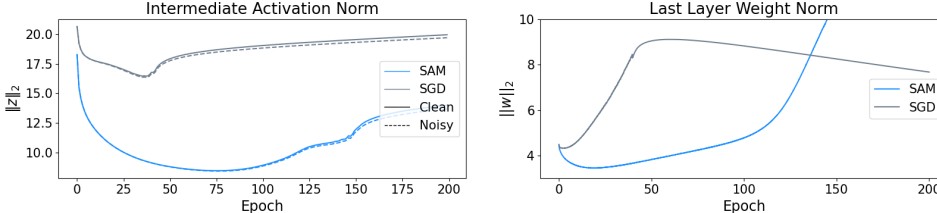

Figure 5: When training ResNet18 with SAM, the norm of the final intermediate activations and last layer weights decreases significantly, consistent with 2-layer DLN analysis.

**Proof** We write the form of the J-SAM update for the first layer $W$ of the deep linear network:

$$-\nabla_{W+\epsilon^{(1)}}\ell(w+\epsilon, x, t) = \sigma(-tf(w,x))\left(tv - \frac{\rho}{J}z\right)x^\top \tag{4.1}$$

$$= -\nabla_W\ell(w,x,t) - \frac{\rho\sigma(-tf(w,x))}{J}zx^\top \tag{4.2}$$

where $z = Wx$ is the intermediate activation and $J = \|\nabla f(x)\| = \sqrt{\|z\|^2 + \|x\|^2\|v\|^2}$ is a normalization factor. In the second layer, the gradient with respect to $v$ is

$$-\nabla_{v+\epsilon^{(2)}}\ell(w+\epsilon, x, t) = \sigma(-tf(w,x))\left(tz - \frac{\rho\|x\|^2}{J}v\right) \tag{4.3}$$

$$= -\nabla_v\ell(w,x,t) - \frac{\rho\sigma(-tf(w,x))\|x\|^2}{J}v \tag{4.4}$$

From Equation 4.1, note that SAM adds an activation norm regularization to the first layer $zx^\top = \nabla_W \frac{1}{2}\|z\|_2^2$ scaled by some scalar dependent on $\rho$, $f(w,x)$, and $J$. Similarly, from Equation 4.3, note that SAM adds a weight norm penalty to the second layer weights $v = \nabla_v \frac{1}{2}\|v\|_2^2$ also multiplied by some scalar. The normalization factor $J$ scales the regularization be closer to the norm than the squared norm. ∎

Empirically, we see that a similar effect is present in SAM for deeper networks. As networks become deeper, the exact J-SAM update becomes more complicated than simply SGD with weight and feature norm regularization. However, when we train ResNet18 with SAM and measure the norm of the final intermediate activations and last layer weights, the norms still decrease significantly in this setting (Figure 5).

## 4.3 EXPERIMENTS

Inspired by our analysis, we train deep networks using SGD with $\ell_2$ regularization on the last layer weights and last hidden layer intermediate activations. Namely, we simplify down SAM's regularization to the following objective

$$\min_w L(w) + \gamma_z \frac{1}{N}\sum_{i=1}^N \|z\|_2 + \gamma_v\|v\|_2^2 \tag{4.5}$$

where $v$ is the last layer weights and $z$ is the last hidden layer intermediate activation.

We conduct our experiments on CIFAR10 with ResNet18. 1-SAM leads to unstable optimization with batch normalization as it requires passing through the datapoints individually through the network. Thus, we replace all batch normalization layers with layer normalization. Keeping all other hyperparameters such as learning rate, weight decay, and batch size the same, we compare the performance of SGD, 1-SAM, L-SAM, J-SAM, and the regularized SGD (Eq. 4.5). Although regularized SGD does not achieve exactly the same test accuracy as SAM, the gap is significantly closed from 17% to 9%. On the other hand, under no label noise, regularized SGD has a neglible improvement of 1% with hyperparameter search, while SAM still achieves 8% higher test accuracy than SGD (Figure 9). Thus, while insufficient to explain all of SAM's generalization benefits, we suspect that a similar regularization of final layers in SAM is particularly important for generalization under heavy label noise.

| Algorithm | Best Test Accuracy |
|---|---|
| 1-SAM ($\rho = 0.01$) | 69.47% |
| L-SAM ($\rho = 0.01$) | 54.13% |
| J-SAM ($\rho = 0.01$) | 69.17% |
| SGD | 52.48% |
| SGD w/ Proposed Reg | 60.8% |

Table 1: Adding our proposed regularization on the last layer weights and logits boosts SGD performance by $8\%$.

### 4.4 CONNECTION BETWEEN LOGIT SCALE AND JACOBIAN TERMS

We hypothesize that the benefits of regularizing the Jacobian could also partially be reasoned about in terms of gradient contribution or as an *implicit* up-weighting scheme. By regularizing the norm of the weights and features, the magnitude of the network output remains small. Given $\|f(x)\|_2 \leq C$, note that the loss of any single datapoint is lower and upper bounded by $\log(1 + (K-1)\exp(-C))$ and $\log(1 + (K-1)\exp(C))$, respectively. By keeping $C$ small, clean training loss may remain non-negligible even as the clean training accuracy increases, and the loss of noisy misfit examples is capped. Indeed, as can be observed in Figure 1 and 7, the loss of clean examples is much higher in SAM than SGD for the same gap between clean and noisy training accuracy. In linear models, weight decay and SAM's logit scale adjustment have equivalent effects (Appendix B). A small weight norm and SAM's logit scale both balance the gradient contribution of examples. Furthermore, weight decay and small initialization is well-known to improve robustness in overparametrized settings (Advani & Saxe, 2017).

## 5 RELATED WORK

### 5.1 IMPLICIT BIAS OF SAM

While the overall mechanics of SAM remains poorly understood, several papers have independently tried to elucidate why the *per-example* regularization of 1-SAM may be important. Andriushchenko & Flammarion (2022a) show that in sparse regression on diagonal linear networks, 1-SAM is more biased towards sparser weights than naive SAM. Wen et al. (2022) differentiate 1-SAM and naive SAM by proving that the exact notion of "flatness" that each algorithm regularizes is different. Our observation of feature regularization is of close connection to Andriushchenko et al. (2023a) which show that SAM drives down the norm of the intermediate activations in a 2 layer ReLU network and this implicitly biases the activations to be low rank. Meng et al. (2023) analyze the per-example gradient norm penalty, which also effectively minimizes sharpness and show that if the data has low signal-to-noise ratio, penalizing the per-example gradient norm dampens the noise allowing more signal learning. Recent works also make connections between naive SAM and generalization (Behdin & Mazumder, 2023; Chen et al., 2024). Our analysis differ from previous works as we focus on understanding 1-SAM's robustness to random label noise, which requires reasoning about the *best not final test accuracy*. Although this is the regime where SAM's achievements are the most notable, it has not been thoroughly studied in previous works which focus on SAM's solution at convergence.

### 5.2 CONNECTION BETWEEN SHARPNESS AND GENERALIZATION

Sharpness of the loss landscape has used in deep learning as a generalization measure (Hochreiter & Schmidhuber, 1997; Dziugaite & Roy, 2017; Bartlett et al., 2017; Neyshabur et al., 2017). A family of optimization choices including minibatch noise in stochastic gradient descent (SGD), large initial learning rate, and dropout have shown to implicitly regularize the model towards solutions lying in *flat* basins or areas of low curvature (Keskar et al., 2016; Dziugaite & Roy, 2017; Cohen et al., 2021; Damian et al., 2022; Nar & Sastry, 2018; Wei et al., 2020; Damian et al., 2021; Orvieto et al., 2022; Jastrzebski et al., 2021). However, the negative correlation between generalization and sharpness is not universally true, but specific to certain optimization choices (Jiang et al., 2019). For example, strong data augmentation may improve generalization but sharpen the landscape (Andriushchenko et al., 2023b). Sharpness regularization also appears in adversarial weight robustness (Wu et al., 2020;

Zheng et al., 2021). However, adversarial weight robustness has no direct connection to the label noise robustness we study in our work.

### 5.3 LEARNING WITH LABEL NOISE

Mislabeled data is a persistent problem even in common benchmarks such as CIFAR and ImageNet (Müller & Markert, 2019) and they can have significant impact on model performance (Nakkiran et al., 2021; Rolnick et al., 2018; Gunasekar et al., 2023; West et al., 2021). Our conclusions, which reduce generalization to how many more clean examples are learned first, is tied to previous literature on the faster learning time of clean versus noisy examples in gradient based optimization. Liu et al. (2020) first prove this in linear models. More recently, Liu et al. (2023) showed that a similar effect is observable in neural networks, where the gradient over clean and noisy examples have negative cosine similarity at the beginning of training, and we may reason about early learning of clean points by the magnitude of their contribution to the average gradient. Many metrics in practice also use learning time to identify examples that are memorized (Zhang & Sabuncu, 2018; Lee et al., 2019; Chen et al., 2019; Huang et al., 2019; Jiang et al., 2017; 2020a; Carlini et al., 2019; Jiang et al., 2020b; Arazo et al., 2019; Liu et al., 2022).

## 6 DISCUSSION, LIMITATIONS, AND CONCLUSION

Although the SAM optimizer has proven very successful in practice, there is a notable divide between the established motivation for SAM of finding a flat minimum, and the empirical gains achieved. Fundamentally, the work we present here aims to justify the usage of SAM by appealing to a very different set of principles than those used to originally derive the algorithm. Specifically, we show that in the linear and nonlinear cases, there is an extent to which SAM "merely" acts by learning more clean examples before fitting noisy examples. This provides a natural perspective upon which to analyze the strong performance of SAM, especially in the setting of label noise.

In the linear setting, we identified that SAM explicitly up-weights the gradient signal from low loss points. This is quite similar to well known label noise robustness methods (Liu et al., 2022; 2020) which also utilize learning time as a proxy for distinguishing between clean and noisy examples. In the nonlinear setting, we identify arguably a more interesting phenomena – *how* clean examples are fit can affect the learning time of noisy examples. We show that SAM regularizes the norm of the intermediate activations and final layer weights *throughout training* and this improves label noise robustness. Ultimately, the effect may be similar to label noise robustness methods that regularize or clip the norm of the logits (Wei et al., 2023).

Finally, we emphasize that despite their close connection, SAM has been surprisingly underexplored in the label noise setting. The research community has developed a number of methods for understanding and adjusting to label noise, and it has so far been a mystery as to how SAM manages to unintentionally match the performance of such methods. Empirically however, we find that simulating even partial aspects of SAM's regularization of the network Jacobian can largely preserve SAM's performance. As a secondary effect of this research, we hope our conclusions can inspire label-noise robustness methods that may ultimately have similar benefits to SAM (but ideally, without the additional runtime cost incurred by sharding the gradients in 1-SAM).

## 7 ACKNOWLEDGEMENTS AND DISCLOSURE OF FUNDING

We thank Hossein Mobahi, Behnam Neyshabur, Dara Bahri, Shankar Krishnan for their insights and guidance which greatly shaped our understanding of SAM. We thank Jeremy Cohen, Tengyu Ma, Maksym Andriushchenko, and Tanya Marwah for the many helpful discussions about sharpness. We thank Jacob Springer, Pratyush Maini, and Alex Li for discussions about memorization, and assistance with building our codebase. We thank Anna Bair and Sanket Mehta for their insights about how SAM behaves in practice. This work was supported in part by Bosch Center for Artificial Intelligence and the AI2050 program at Schmidt Sciences (Grant #G2264481).

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

# A    METHODS

All code was implemented in JAX (Bradbury et al., 2018), and we utilize the Flax neural network library. We utilize a normally distributed random initialization scheme. Our experiments were run on NVIDIA Quadro RTX A6000.

## A.1    SYNTHETIC LABEL NOISE

We add synthetic label noise in the following manner. We randomly select $\Delta$ proportion of the training data to corrupt. For each datapoint $(x, y)$, we select corrupted label $t$ randomly from $t \in \{i \in [k]_{k=1}^K \mid i \neq y\}$.

## A.2    EXPERIMENTS ON CIFAR10

For experiments conducted on CIFAR10, we train with $30\%$ label corruption. We hyperparameter tuned the model for the best learning rate using SGD, and then hyperparameter tune 1-SAM's hyperparameter $\rho$ utilizing the same learning rate. We do not utilize any learning rate schedule or data augmentation other than normalizing the image channels using mean (0.4914, 0.4822, 0.4465) and standard deviation (0.2023, 0.1994, 0.2010).

**ResNet18**    We have modified the ResNet18 architecture by replacing all Batch Norm layers with Layer Norm. This is necessary to safely run 1-SAM which requires a separate forward pass for each datapoint in the minibatch.

| Parameter | Value |
|---|---|
| Batch size | 128 |
| Learning rate | 0.01 |
| Weight decay | 0.0005 |
| Epochs | 200 |
| $\rho$ (for SAM) | 0.01 |

**2 Layer DLN/MLP with ReLU**    We do not include bias at any layer. The width of the intermediate layer is set to 500. We use the same hyperparameters as the ResNet18 experiments.

## A.3    EXPERIMENTS ON TOY DATA

**Linear**    We set the parameters of our toy data distribution to be the following

| Parameter | Value |
|---|---|
| $\Delta$ | 0.4 |
| B | 2 |
| $\gamma$ | 1 |
| d | 1000 |
| Training samples | 500 |
| Test samples | 1000 |

There is no weight decay. Learning rate is set to $0.01$.

# B LINEAR MODEL ANALYSIS

**Toy Data Setting** We train our linear models over the following toy Gaussian data distribution $\mathcal{D} : \mathbb{R}^d \times \{-1, 1\}$.

$$\text{True Label: } y \sim \{-1, 1\} \text{ by flipping fair coin}$$

$$\text{Input: } x \sim y \cdot \left[ B \in \mathbb{R}, z \sim \mathcal{N}\left(0, \frac{\gamma^2}{d-1}I_{d-1\times d-1}\right) \right] \tag{B.1}$$

$$\text{Target: } t = y \cdot \varepsilon \text{ where } \varepsilon \sim \{-1 \text{ w.p } \Delta, \; 1 \text{ w.p } (1-\Delta)\}$$

The test data is generated from a related distribution where the target is noiseless $t = y$.

**Test Accuracy** The expected accuracy over the test distribution can be written as

$$\text{Acc}(w) = \mathbb{E}_{x,y\sim\mathcal{D}_{test}}\left[ \mathbb{1}\left[ y(w^\top x) > 0 \right] \right] = P\left( y(w^\top x) > 0 \right) \tag{B.2}$$

$$= P_{z\sim\mathcal{N}(0,I)}\left( \frac{\gamma}{\sqrt{d-1}}w_{1+}^\top z > -w_1 B \right) = 1 - \Phi\left( -\frac{B\sqrt{d-1}w_1}{\gamma\|w_{1+}\|} \right) \tag{B.3}$$

where $w_{1+} \in \mathbb{R}^{d-2}$ denotes the vector consisting of the entries in $w$ excluding the first. Therefore, the accuracy monotonically increases with $w_1/\|w_{1+}\|$ and optimal linear classifier in this setting is proportional to the first elementary vector:

$$w^* \propto e_1 \tag{B.4}$$

## B.1 1-SAM ASYMPTOTICS

We are interested in assessing the early-stopping test accuracy of naive SAM (n-SAM) versus 1-SAM. As we can observe in Figure 3, the best test accuracy monotonically increases with $\rho$. We analyze 1-SAM in the limit as the perturbation magnitude converges to limit $\rho \to \infty$ and observe that it converges to the optimal classifier. In this regime, the 1-SAM update for each example simply becomes

$$\nabla_{w+\varepsilon_i}\ell(w + \varepsilon_i, x_i, y_i) = \lim_{\rho\to\infty} -t_i\sigma\left(-t_i\langle w_i, x_i\rangle + \rho\|x_i\|_2\right)x_i = -t_i x_i \tag{B.5}$$

and the ratio between the magnitude of any two points converges to 1, specifically for any two datapoints $x_i$ and $x_j$

$$\lim_{\rho\to\infty} \frac{\|\nabla_{w+\varepsilon_i}\ell(w+\varepsilon_i, x_i, y_i)\|}{\|\nabla_{w+\varepsilon_j}\ell(w+\varepsilon_j, x_j, y_j)\|} = \lim_{\rho\to\infty} \frac{\sigma\left(-t_i\langle w, x_i\rangle + \rho\|x_i\|\right)}{\sigma\left(-t_j\langle w, x_j\rangle + \rho\|x_j\|\right)} \tag{B.6}$$

$$= \lim_{\rho\to\infty} \frac{1 + \exp(t_i\langle w, x_i\rangle - \rho\|x_i\|)}{1 + \exp(t_j\langle w, x_j\rangle - \rho\|x_j\|)} = 1 \tag{B.7}$$

As a result, each gradient update of 1-SAM is precisely equal the empirical mean of the data scaled by the label $\hat{u}_{\mathcal{D}} = X^\top t$. Say that we are given fixed training examples $[x_i]_{i=1}^n$ independently sampled from $\mathcal{D}$ where $\Delta$ is corrupted. Note that as $d$ or $n$ grows (and $\Delta < 0.5$), $\hat{u}_{\mathcal{D}}$ converges to the Bayes optimal classifier. Notably,

$$\hat{u}_{\mathcal{D}} = \left[ (1-2\Delta)B, \sum_{i=1}^n \frac{\gamma}{n\sqrt{d-1}}z_i \right] \xrightarrow{d,n\to\infty} (1-2\Delta)Be_1 \tag{B.8}$$

Note that the ridge regression also converges to the empirical mean scaled by the label in the limit

$$(XX^\top + \lambda I)^{-1}X^\top t \xrightarrow{\lambda\to\infty} X^\top t \tag{B.9}$$

This gives us reason to believe that SAM and weight decay have similar regulatory properties.

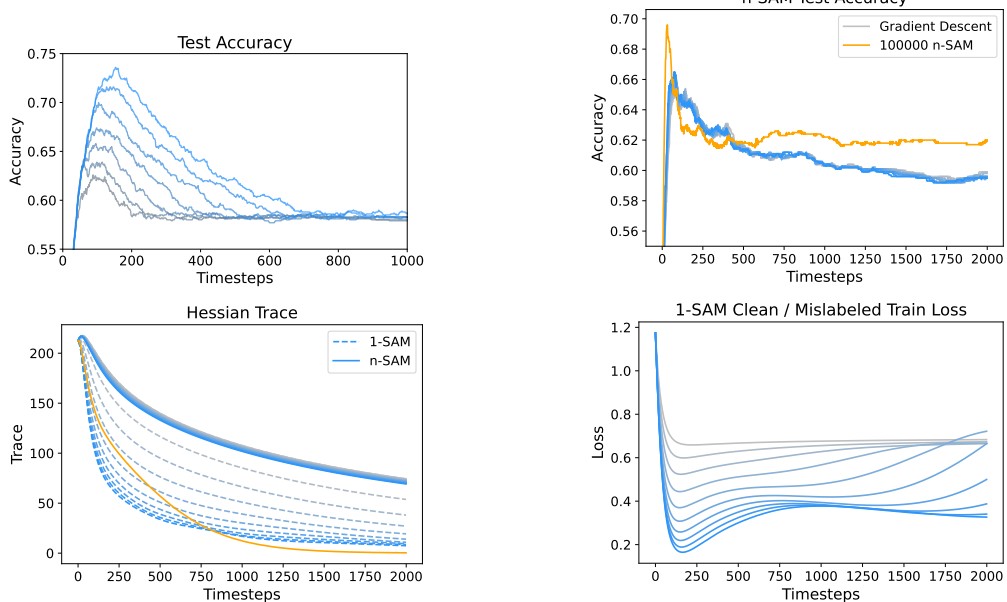

Figure 6: We train linear models on our toy model using 1-SAM. Bluer curves denote larger $\rho$. We randomly initialize the model, although we observe similar curves when initialized at the origin.

## B.2 ANALYSIS OF N-SAM

We also analyze the naive SAM (n-SAM) update

$$w = w - \eta \left( \frac{1}{n} \sum_{i=1}^{n} \nabla_{w+\epsilon} \ell \left( w + \epsilon, x_i, y_i \right) + \lambda w \right) \tag{B.10}$$

where $\epsilon = \rho \frac{\nabla_w L(w)}{\|\nabla_w L(w)\|_2}$ and $L(w) = \frac{1}{n} \sum_{i=1}^{n} \ell(w, x_i, y_i)$ and $\rho$ is a hyperparameter. The original and follow up works Foret et al. (2020); Andriushchenko & Flammarion (2022a) show that n-SAM does not have any advantage over gradient descent in practice. We observe that even in the linear setting with toy Gaussian data, n-SAM observes little early stopping improvements unless $\rho$ is scaled up dramatically. In Figure 4 for example, we set $\rho = 100000$ for n-SAM.

Comparing 1-SAM and n-SAM in the linear setting, we find that they have fundamentally different effects. n-SAM's perturbation is not a constant that is only a function of the data norm. The magnitude of the perturbation is proportional to the loss, so SAM does not preferentially upweight low loss points except for the early training steps. Let us consider that the noise $z_i$ of the datapoints are orthogonal for ease of analysis. Then

$$\nabla_w^{n-SAM} \ell(w, x_i, y_i) = -t_i \sigma \left( -t_i \langle w + \rho \frac{\nabla_w L(w)}{\|\nabla_w L(w)\|}, x_i \rangle \right) x_i \tag{B.11}$$

$$= -t_i \sigma \left( -t_i \langle w, x_i \rangle + \frac{\rho}{n \|\nabla_w L(w)\|} \left( CB^2 + \frac{\gamma^2 \|z_i\| \sigma(-t_i \langle w, x_i \rangle)}{d-1} \right) \right) x_i \tag{B.12}$$

for some scalar C that remains constant across the examples. Assuming gradient descent starting at $w = 0$, class balance, and the same number of mislabeled datapoints in each class, n-SAM at each iteration, perturbs each point proportional to $\sigma(-t \langle w, x_i \rangle)$ which is smaller for low loss points and higher for high loss points. This does not guarantee preferential up-weighting. We do observe in Figure 3 (see orange curve) that if $\rho$ is sufficiently large, we are able to see some gains with n-SAM in the first couple timesteps. Finally, we do not see any correlation between test accuracy and flatness measured by the Hessian trace. In fact, n-SAM with large $\rho$ achieves smaller Hessian trace by the end of training but lower test accuracy.

## C    IMPLICIT UP-WEIGHTING

Previously, we showed that the gap between the training accuracy of clean and noisy examples correlates highly with the test accuracy (See Figure 7). This led us to reason about improved generalization with learning clean examples faster than noisy examples. In particular, we use the "sample-wise update norm" as a proxy for how much each clean and noisy example contribute to the average update. For example, with SGD, the update is the gradient evaluated at the model parameters $w$, and for SAM, the update is the gradient evaluated at the perturbed $w + \epsilon$. We showed that in the linear setting, the logit scale term of SAM up-weights the update norm of clean examples. Next, for non-linear setting, we showed that SAM's Jacobian term regularizes the magnitude of the function output, and we suspect that this has a similar implicit effect of balancing the gradients of clean and noisy examples as the loss of clean examples decreases first.

Below, we plot the average clean versus noisy ratio of the loss, sample-wise update norm, and logit scale norm from Section 4.1. We desire these quantities to be high even as the clean-noisy accuracy gap increases. For the same clean-noisy accuracy gap, SAM and J-SAM's ratio of clean/noisy loss is higher (across the training trajectory, before the accuracy gap peaks). As a result, the clean/noisy update norm ratio is also higher with SAM and J-SAM, meaning clean examples have higher influence on the direction of the update . Notably, this up-weighting effect is implicit, not due to explicit upweighting.

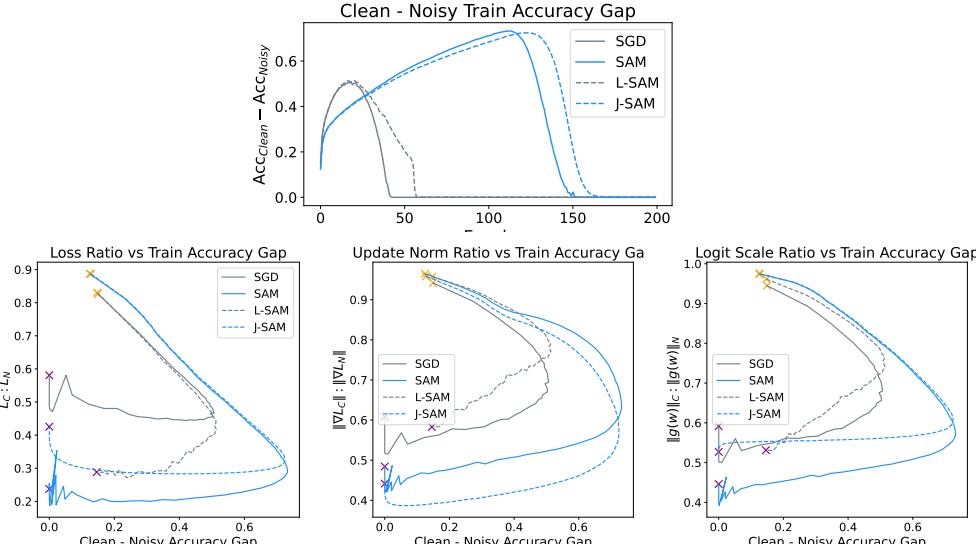

Figure 7: We plot the clean/noisy ratio of the loss, sample-wise gradient norm, and logit scale norm versus the clean-noisy accuracy gap. The ratios are higher with SAM and J-SAM than SGD and L-SAM for the same accuracy gap. Orange marker denotes the beginning of the training trajectory and purple denotes the end.

## D EVIDENCE OF LABEL NOISE ROBUSTNESS ON OTHER DATASETS

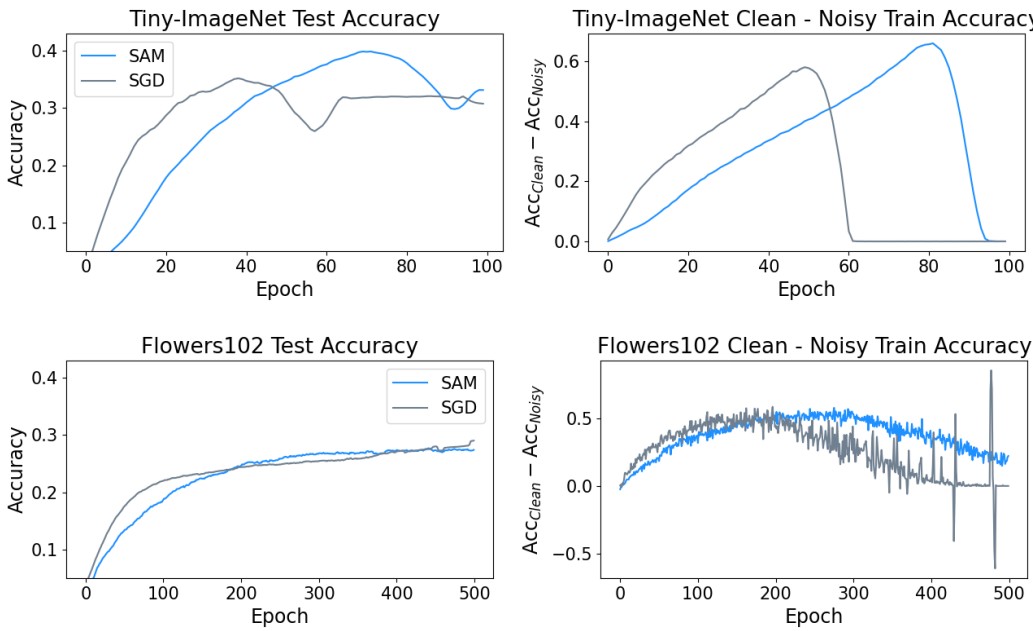

Figure 8: SAM observes higher true accuracy of noisy examples and this corresponds with better test accuracy on Tiny-ImageNet with $30\%$ label noise and Flower102 with $20\%$ label noise. Do note that contrary to trends in CIFAR10, in a low data settings such as Flowers102, the test accuracies of SAM and SGD do not drop even when the model starts to overfit. Models are ResNet18.

## E REGULARIZED SGD UNDER NO LABEL NOISE

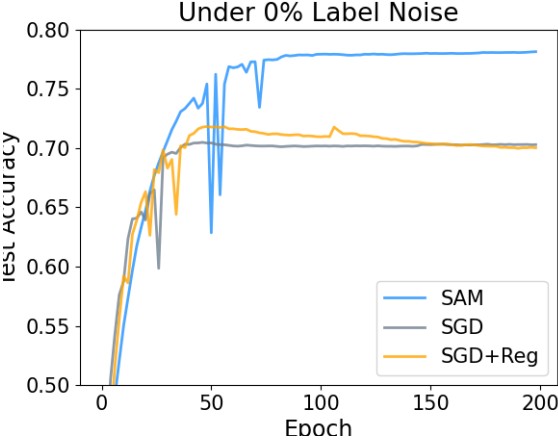

Figure 9: SAM ($\rho = 0.01$) and SGD have a smaller $8\%$ difference in performance (less in comparison to the $20\%$ difference with $30\%$ label noise). Our weight and feature norm penalty observes improvements but only by a small factor of $1\%$ and the performance degrades over time.

# F  ABLATION STUDIES

We observe that the improved benefits of SAM in the label noise setting does not dimish with underparametrization. Furthermore, SGD with large learning rate cannot match SAM's performance.

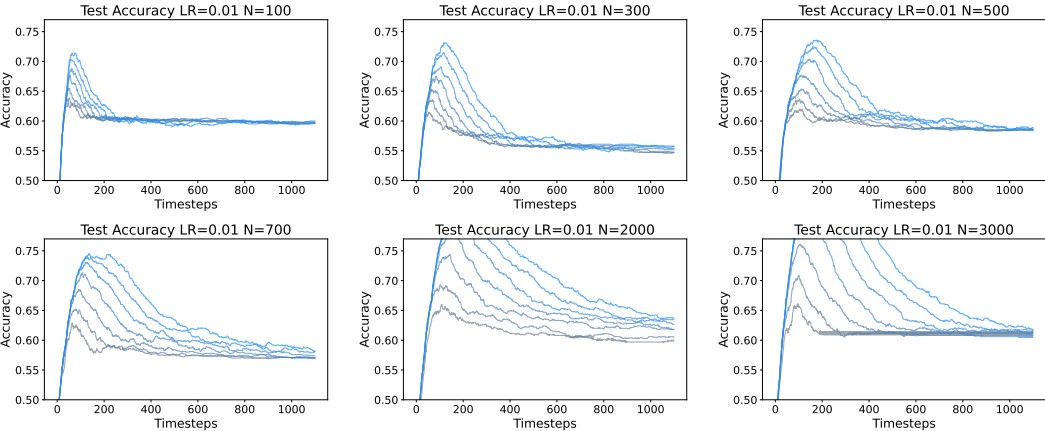

Figure 10: **Behavior with number of training examples for linear model/toy Gaussian data with 40% label noise.** Linear models trained with different subsets of the toy Gaussian data observe different levels of benefit with SAM. $\rho$ is scaled between 0.03 and 0.18, bluer curves signifying higher $\rho$. The data is 1000 dimensions. Note that performance improves with SAM even in underparametrized regimes.

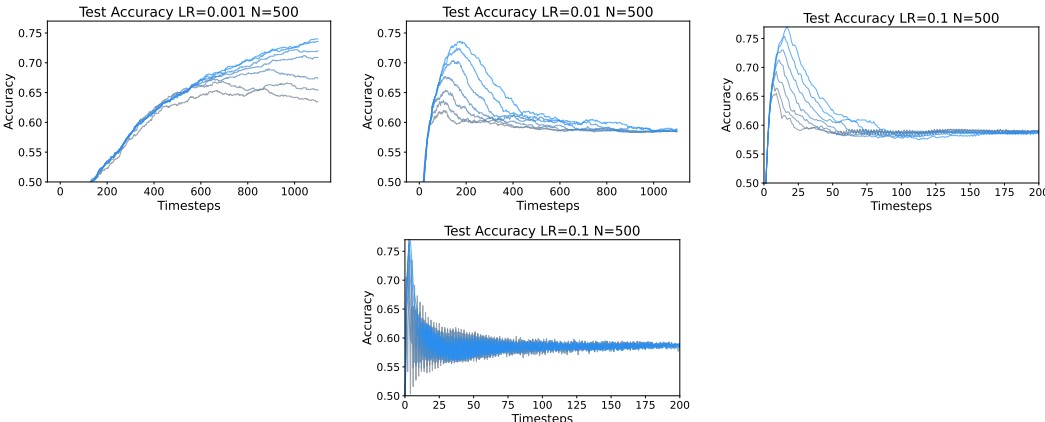

Figure 11: **Behavior with learning rate for linear model/toy Gaussian data with 40% label noise** As the learning rate increases, we generally observe a slight improvement in early stopping accuracy in both SGD and SAM.

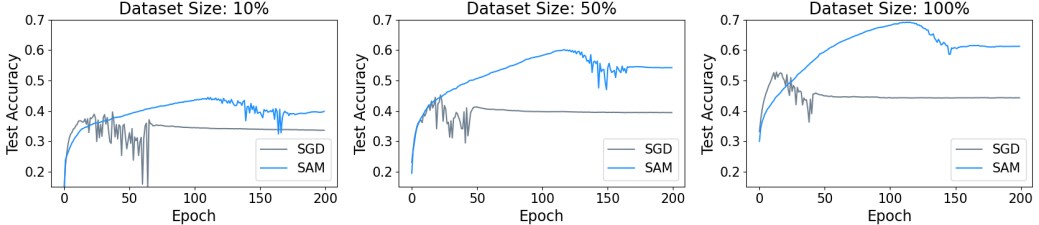

Figure 12: **Behavior with number of training examples for ResNet18/CIFAR10 with 30% label noise** We compare SGD and SAM trained on different number of training examples $(10, 50, 100\%$ of the training data). We see that the difference between SAM $(\rho = 0.01)$ and SGD increases as the dataset size increases.

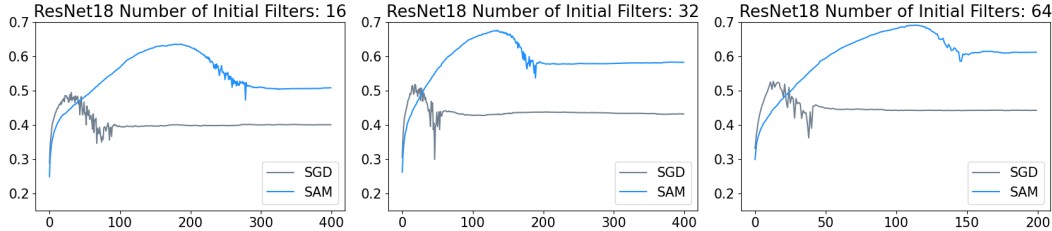

Figure 13: **Behavior with model width for ResNet18/CIFAR10 with 30% label noise** ResNet18 starts with 64 convolutional filters, and the filters double every two convolutional layers. We reduce the width of ResNet18 by 1/2 and 1/4 by starting with 32 and 16 initial number of filters, respectively. We see that SAM $(\rho = 0.01)$ and SGD both improve in terms of performance as model width increases.

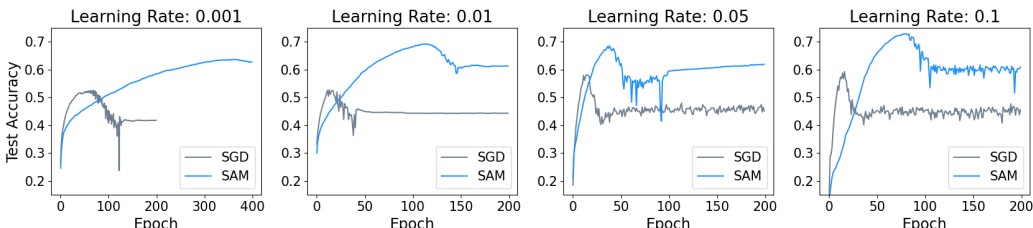

Figure 14: **Behavior with learning rate for ResNet18/CIFAR10 with 30% label noise**. For each learning rate, we choose the best $\rho$ for SAM found by hyperparameter search. As learning rate increases, we observe that both SAM and SGD both improve in terms of performance as learning rate increases. For small learning rate 0.001, we found it difficult for SAM to observe significant improvements upon SGD unless trained for at least double the time.

