# OpenReview forum: "Why is SAM Robust to Label Noise?"
_ICLR.cc/2024/Conference — ICLR 2024 poster_

### Official Review · Reviewer_Cbxh · 2023-10-27

**Soundness:** 3 good
**Presentation:** 2 fair
**Contribution:** 2 fair
**Rating:** 6
**Confidence:** 4

**Summary:**

This paper analyzes the label noise robustness of the SAM (Sharpness-Aware Minimization) optimizer. SAM is known to achieve large gains in test accuracy over SGD when there is label noise, but the reasons are not well understood. The authors decompose SAM's sample-wise gradient into a logit term and Jacobian term. In linear models, they show SAM's logit term acts like an explicit reweighting that upweights low-loss (likely clean) examples. However, in neural networks, SAM's gains come primarily from regularization effects induced by the Jacobian term rather than explicit reweighting. The authors analyze the Jacobian term in a 2-layer linear network, showing it induces feature norm regularization. Adding just these implicit regularization terms recovers much of SAM's performance.

**Strengths:**

1. Provides theoretical analysis and experiments investigating an important practical phenomenon - SAM's label noise robustness.

2. Careful decomposition and ablation studies (logit vs Jacobian SAM) elucidate the source of gains.

3. Analysis of the Jacobian term shows it induces implicit regularization that aids robustness.

4. Proposes a simplified method motivated by analysis that recovers much of SAM's gains.

**Weaknesses:**

1. Analysis limited to 2-layer linear networks, unclear if insights extend to deep nonlinear networks.

2. Lacks comparison to other label noise robust methods.

**Questions:**

Does the analysis for 2-layer linear networks provide insights into deep nonlinear networks? What are the limitations?

Could you compare the proposed simplified method to existing techniques like MentorNet?

---

> ### Author Response · Authors · 2023-11-17
> **Thank you for your time and constructive feedback**
>
> Thank you for the feedback! We appreciate your recognition of the strengths of our paper. We address your concerns below.
>
> **1. Analysis limited to 2-layer linear networks, unclear if insights extend to deep nonlinear networks.**
>
> Great question! To reiterate, our nonlinear section makes the following connection between the 2-layer linear network analysis and deep networks:
> - In 2-layer linear networks, SAM regularizes the norm of the intermediate features and last layer weights (Equation 4.6).
> - If we apply a feature regularization and weight decay in a similar fashion to the _last two layers_ of a deep model (ResNet18), we can close the gap between SAM and SGD halfway.
>
> We’ve also added the following new experiment to further connect the two.
>
> - When training with SAM, the norm of the last-layer activations and weights are smaller than SGD. In Figure 7 (Page 19), we plot the norm of the last-layer activations and weights during the training trajectory of SAM and it is indeed noticeably smaller.
>
>     This suggests that SAM does implicit feature/weight norm regularization in deep models, similar to what we identified in the 2-layer linear network analysis, and this regularization is important for label noise.
> - Under no label noise, we find that our regularization method (with hyperparameter search) only leads to a 1% boost in accuracy and the gain disappears at convergence. See Figure 5 (Page 18) for the test accuracy plots for SAM, SGD, and SGD + our weight/feature penalty.
> \
> \
>     While under label noise, the last layer regularization closes half the SAM-SGD performance gap, the regularizer only closes an eighth of the performance gap with no label noise. This finding suggests that last-layer feature regularization, which is clearly only just a small aspect of SAM, is especially important under label noise.
>
> **2. Lacks comparison to other label noise robust methods. Could you compare the proposed simplified method to existing techniques**
>
> Thanks for the suggestion! We’d like to first emphasize that this paper is not proposing any state-of-the-art label noise robustness technique. Instead, we would like to understand why SAM, an optimizer designed to converge to flat minima, has surprisingly good label noise robustness close to state-of-the-art.
>
> Previous works do a thorough comparison of SAM to other state-of-the-art methods:
> - Table 4 of [1] compares SAM against methods including Co-teaching, MentorNet, Mixup, MentorMix, and Bootstrap. They observe SAM is better than these methods by 0-5% on CIFAR10.
> - Table 9 of [2] compares SAM against logit clipping, and shows SAM can benefit from additional logit clipping by 3%.
>
> In our paper, we restrict comparing the performance of our regularizer to SAM since our goal is not to achieve state-of-the-art but understand what makes SAM robust. We see that the penalty does _halve the gap between SAM and SGD’s performance_ suggesting that feature regularization at the last layer can be a key component for explaining why SAM is robust to label noise.
>
> We hope these adjustments address your main concerns. Please let us know if you have any further feedback!
>
> **References**
>
> [1] Foret et al (2020) Sharpness-Aware Minimization for Efficiently Improving Generalization [https://arxiv.org/abs/2010.01412]
>
> [2] Wei et al (2022) Mitigating Memorization of Noisy Labels by Clipping the Model Prediction  [https://arxiv.org/abs/2212.04055]

---

> > ### Author Response · Authors · 2023-11-21
> > **Any further questions?**
> >
> > Thank you again for the comprehensive and useful review! According to your suggestions, we've
> >
> > 1. **Strengthened the connection between 2-layer DLN analysis and deeper models**: Training with SAM on deep models also shows feature and weight regularization. Under no label noise, feature regularization doesn't lead to as much gain.
> > 2. **Compared SAM to other robust training algorithms**: Although our work focuses on understanding SAM in particular, previous works show SAM have close performance to other robust training algorithms and have benefits on top of them.
> >
> > Does this address your concerns? If you have any more questions or feedback, please let us know.

---

> > > ### Comment · Reviewer_Cbxh · 2023-11-23
> > > **Official Comment by Reviewer Cbxh**
> > >
> > > Thank you for the responses.  They provide clarification on the questions raised.  After discussing with other reviewers, I will reassess my score.

---

### Official Review · Reviewer_WpDM · 2023-10-28

**Soundness:** 3 good
**Presentation:** 3 good
**Contribution:** 3 good
**Rating:** 6
**Confidence:** 4

**Summary:**

This paper examines why SAM has better generalization performance than SGD in the presence of label noise. This phenomenon can't be explained by flatness minimization because the best performance is usually reached before the loss converges. The author decomposed SAM's robustness into two effects, one induced by the logit term and the other induced by changing network Jacobian. In the linear setting, the Jacobian is independent of weight, and the logit effect upweights the gradient of clean examples. In a neural network setting, however, the logit effect is neither necessary nor sufficient for performance improvement. The authors conclude by deriving a regularization method that is cheaper than SAM and can almost recover the benefit of SAM for experiments on CIFAR10.

**Strengths:**

* **Originality.** Although the robustness of SAM towards label noise has been discussed, this paper shows surprisingly logit effect is in fact not important for this robustness.

* **Clarity.** The paper is well-written and easy to read.

* **Significance.** The paper examines an interesting and important question in understanding SAM.

**Weaknesses:**

* Equation 4.5 includes a stop gradient operator in a minimization target, which, to the reviewer's knowledge, is a non-standard way of writing. The reviewer would recommend to rephrase into an update rule.

**Questions:**

* How would the regularization method perform when there is no label noise present?

* Is the performance gain bring by SAM additive to current robust training algorithm or will this performance gain diminishes when more sophisticated training algorithm than SGD is used?

---

> ### Author Response · Authors · 2023-11-17
> **Thank you for your time and constructive feedback**
>
> Thank you for the feedback! We appreciate your recognition of the strengths of our paper. We address your concerns below.
>
> **1. Equation 4.5 includes a stop gradient operator in a minimization target, which, to the reviewer's knowledge, is a non-standard way of writing. The reviewer would recommend rephrasing into an update rule.**
>
> Thank you for the suggestion, we’ve modified the paper accordingly.
>
> **2. How would the regularization method perform when there is no label noise present?**
>
> Under no label noise, we find that our regularization method (with hyperparameter search) only leads to a 1% boost in accuracy and the gain disappears at convergence. See Figure 5 (Page 18) for the test accuracy plots for SAM, SGD, and SGD with our regularization.
>
> While under label noise, the last layer regularization closes half the SAM-SGD performance gap, the regularizer only closes an eighth of the performance gap with no label noise. This finding suggests that last-layer feature regularization, which is clearly only just a small aspect of SAM, is especially important under label noise.
>
>
> **3. Is the performance gain brought by SAM additive to current robust training algorithms or will this performance gain diminish when a more sophisticated training algorithm than SGD is used?**
>
> Great question! The analysis we provide for SAM in the linear section shows that in SAM has properties similar to other label noise robustness methods that also leverage the speed at which examples are learned i.e. clean points losses tend to go down faster than noisy point losses (see our Related Works). In direct relation are methods that prevent gradient starvation of low loss points either by logit clipping [1] or a more balanced loss function than cross-entropy [2].
>
> Getting back to your original question, when applying SAM on top of label noise robustness methods, it may be possible to observe further improvements. See Table 4 in [3] where SAM observes further gains with bootstrapping and Table 9 of [1] where SAM is paired with logit clipping.
>
> We hope these adjustments address your main concerns. Please let us know if you have any further feedback!
>
> **References**
>
> [1] Wei et al. (2022) Mitigating Memorization of Noisy Labels by Clipping the Model Prediction [https://arxiv.org/abs/2212.04055]
>
> [2] Ghosh et al. (2017) Robust Loss Functions under Label Noise for Deep Neural Networks [https://arxiv.org/abs/1712.09482]
>
> [3] Foret et al. (2020) Sharpness-Aware Minimization for Efficiently Improving Generalization  [https://arxiv.org/abs/2010.01412]

---

> > ### Author Response · Authors · 2023-11-21
> > **Any further questions?**
> >
> > Thank you again for the comprehensive and useful review! According to your suggestions, we've
> >
> > 1. **Strengthened our nonlinear section**: Provided further results that indicate feature regularization is especially important under label noise. Without label noise, feature regularization doesn't help very much.
> > 2. **Compared SAM to other robust training algorithms**: Although our work focuses on understanding SAM in particular, previous works show SAM may have benefits on top of other robust training algorithms. We also discuss similarities in methodology.
> >
> > Does this address your concerns? If you have any more questions or feedback, please let us know.

---

> > > ### Comment · Reviewer_WpDM · 2023-11-22
> > >
> > > Thank you for the response. It answers my questions and I will keep my score.

---

### Official Review · Reviewer_M3rX · 2023-11-01

**Soundness:** 2 fair
**Presentation:** 3 good
**Contribution:** 2 fair
**Rating:** 6
**Confidence:** 3

**Summary:**

This paper provides analysis to understand robustness of SAM to label noise through the lens of implicit regularization. The key idea is that the benefits of SAM can be primarily attributed to the network Jacobian part appearing in the sample-wise gradient. Analysis of the Jacobian term is then provided in simplified settings and empirical experiments are also provided to illustrate the general applicability of the idea (in CIFAR-10 classification).

**Strengths:**

- Provide refreshing insights on robustness of SAM to input labels through the lens of implicit regularization
- Overall the paper is well written and is easy to follow

**Weaknesses:**

- No analysis/empirical demonstrations on tasks other than classification are provided (e.g., regression tasks)
- Missing discussions/analysis on how the robustness benefits depend on parameters such as number of parameters, number of training samples, learning rate , etc. (see also Questions below)
- Missing some references in Related Work, e.g.: https://arxiv.org/abs/1609.04836, https://arxiv.org/abs/1705.10694

**Questions:**

- How does robustness of SAM depend on the network width/number of parameters (d) and number of training samples (n)? Are there additional benefits (or otherwise) that SAM provide in the overparametrized regime (or some non-trivial regimes in terms of n and d)?
- How does robustness of SAM in the stage of SGD training depends on the learning rate? Does the learning rate need to be small enough  to better isolate the benefits of SAM?
- Perhaps one could investigate the above  questions in the setting of Section 3.1 and also perform empirical studies on benchmark tasks like CIFAR-10 classification?


Minor comments:
- typos in the formula for $\epsilon_i$ in Eq. (2.6): $y_i \mapsto t_i$

---

> ### Author Response · Authors · 2023-11-17
> **Thank you for your time and constructive feedback**
>
> Thank you for the constructive feedback! We appreciate your recognition of the strengths of our paper. We address your concerns below.
>
> **1. No analysis/empirical demonstrations on tasks other than classification are provided (e.g., regression tasks)**
>
> This is correct, we focus on understanding SAM’s regularization for the cross-entropy loss (classification), which may have different qualitative behaviors than SAM for the squared loss (regression). We hope this is clear from our abstract, and we believe our findings for classification alone can still be quite interesting to the research community.
>
> **2. Missing some references in Related Work, e.g.: https://arxiv.org/abs/1609.04836, https://arxiv.org/abs/1705.10694**
>
> Thank you for the pointers to these works! Keskar was already cited in Subsection 5.2, and we added Rolnick to Subsection 5.3 of our related works section.
>
> **3. How does robustness of SAM depend on the network width/number of parameters (d) and number of training samples (n)? Are there additional benefits (or otherwise) that SAM provide in the overparameterized regime (or some non-trivial regimes in terms of n and d)?**
>
> Great questions! To answer, we added the following ablation studies on Page 19 to observe the effect of width (or overparameterization) on the performance difference between SAM and SGD.
> - Figure 9 (Page 19), SAM vs SGD in linear models trained on different numbers of examples.
> - Figure 11 (Page 20) SAM versus SGD trained on _different data percentages_ of CIFAR10 on ResNet18.
> - Figure 12 (Page 20) SAM versus SGD on CIFAR10 for _different number of convolution filters_ (to adjust for width) in ResNet18.
>
> We observe that in both linear and nonlinear models, for the same SAM perturbation rho, the gap between SAM and SGD actually increases with the number of training examples. In particular, looking at the effect in linear models in Figure 9, one can observe that the difference between SAM and SGD is especially significant when the model is under-parametrized ($n > d$).
>
> This may seem contrary to your expectations. However, note that we are not analyzing the performance of models at convergence, but the best test performance along its training trajectory. As a result, even when the loss is strictly convex where SAM and SGD may _converge_ to the same solution, depending on the training trajectory, the early stopping accuracy can be very different between the two optimizers.
>
> On the other hand, we see in Figure 12 that while decreasing the number of data (keeping the model size fixed) makes the model more “overparameterized”, but _reduces_ the gap between SAM and SGD, overparameterization by increasing model width (and fixing data size) _increases_ the gap between SAM and SGD. We suspect that the relationship between label noise and overparameterization/width is complicated, and do not draw any strong conclusions from these results.
>
> **4. How does robustness of SAM in the stage of SGD training depends on the learning rate? Does the learning rate need to be small enough to better isolate the benefits of SAM?**
>
> Great question! See Figure 10 (Page 20) and 13 (Page 21), where we conduct an ablation study over the learning rate for SGD and SAM in linear and nonlinear models respectively. We do a grid search over SAM’s hyperparameter rho for each learning rate. Indeed as the learning rate scales up, SGD’s early stopping test accuracy also increases. However, SAM’s performance also improves and
> SAM’s gains do not disappear at higher learning rates.
>
> **5. Typos in the formula 2.6**
> Thanks! We fixed this.
>
> We hope these adjustments address your main concerns. Please let us know if you have any further feedback!

---

> > ### Author Response · Authors · 2023-11-21
> > **Any further questions?**
> >
> > Thank you again for the comprehensive and useful review! According to your suggestions, we've provided ablation studies looking at the effect of network **width**, **number of data**, and **learning rate**. We saw larger learning rate improves performance of both SAM and SGD, and too little data diminishes the gap between the best SAM and best SGD performance.
> >
> > Does this address your concerns? If you have any more questions or feedback, please let us know.

---

> > > ### Comment · Reviewer_M3rX · 2023-11-22
> > > **Thank you for addressing my concerns**
> > >
> > > I appreciate the efforts of the authors in addressing my concerns and I am happy to see these discussions being included in the revised version.

---

### Official Review · Reviewer_ATLV · 2023-11-07

**Soundness:** 3 good
**Presentation:** 3 good
**Contribution:** 3 good
**Rating:** 6
**Confidence:** 3

**Summary:**

The submission studies early stopping performance of Sharpness-Aware Minimization (SAM) under label noise. The effect of SAM on optimization is first decomposed into a logit term and a Jacobian term. In logistic regression, the Jacobian term is ineffectual and the effect is totally explained by the logit term which upweights the gradients of clean labels and delays fitting the noise. In neural networks, the logit term plays a similar role of reweighting gradients. However, here this term has little effect on the overall performance and the beneficial effects are due to the Jacobian term. A simple theoretical analysis on a two-layer linear network shows that the Jacobian term regularizes the representation and the last layer weights.

**Strengths:**

Understanding the effect of SAM is of paramount interest due to the popularity of this technique. The baselines and the experiments are designed to directly answer the questions. The theory, although rather simple, is not known nor trivial. The related work is adequately covered.

**Weaknesses:**

The following concerns are the reasons for the low score and I can raise the score if all three are addressed.

**1. Little evidence on the role of early stopping.** Most of the narrative highlights that SAM is especially effective when combined with early stopping. The importance of early stopping in the analysis is emphasized throughout the paper. However, when I look at the ResNet experiments in Fig 1 and 3, early stopping seems to have little to no effect, and the difference in performance is already largely present in the final stage of training. I ask the authors to either provide more evidence on the special role of early stopping or edit the text in the abstract, introduction, and sections 5 and 6 to deemphasize the importance of early stopping. In addition, the presentation of the middle plot in Figure 1 is problematic: The caption says "SAM fits much more clean data however before beginning to overfit to mislabeled data" but the evidence is hard to infer from the plot. The revision should present this result more clearly.

**2. Little insight on the effects of the regularization.** Section 4.2 shows that the role of the Jacobian term is similar to a certain regularization on the representation and the final layer weights. The discussion does not properly connect this regularization effect to the overall narrative about robustness to label noise and the role of early stopping. The text below the theory only briefly says "In linear models, weight decay has somewhat equivalent effects to SAM’s logit scaling in the sense that it balances the contribution of the sample-wise gradients and thus, prevents overfitting to outliers," but I did not find any basis for this claim, nor any discussion on the effect of regularizing the representation.

**3. Inadequate empirical support.** The large-scale experiments in the submission are only on Cifar-10. This is not nearly enough for an ICLR publication and hardly supports the claims in the paper. There are many other medium- to large-scale datasets (Tiny ImageNet, ImageNet, MS-COCO, flowers102, places365, food101, etc.) and the revision should include at least one of these datasets (the new dataset should not be too similar to Cifar-10 like Cifar-100 or too small like MNIST).

Other comments:
- In regression there is a rigorous theoretical framework for studying the role of early stopping on performance under label noise [1,2]. The type of task and label noise in this framework is different from the submission and discussing these tools is outside the scope of this paper but the authors may find them interesting for future work.

[1] Advani, Madhu S., Andrew M. Saxe, and Haim Sompolinsky. "High-dimensional dynamics of generalization error in neural networks." Neural Networks 2020.

[2] Ali, Alnur, J. Zico Kolter, and Ryan J. Tibshirani. "A continuous-time view of early stopping for least squares regression." AISTATS 2019.

-----
Post-Rebuttal: I raised the score to accept since the original comments have been addressed. Here are a few minor comments for the final version:
- Section 2.1: The first line defines the input space as R^d and the second line as $\mathcal{X}$. If the input is always a d-dimensional vector then there is no need for the extra notation $\mathcal{X}$.
- Eq 2.1: It seems to me that W is an h by d matrix
- The new plot in Fig. 1 is a good addition. It's better to reorganize this figure now the reduce the empty space.
- There is still some extra space at the end of the paper. Bringing some of the results from the appendix to the main paper would be helpful.
- Eqs 4.1 and 4.3: The derivation of the Jacobian term is not obvious to me. I suggest laying out the one or two intermediate steps either in the main paper or in the appendix. Also, the left-hand side uses a gradient notation even though J-SAM is not the true gradient.

**Questions:**

See Weaknesses 1, 2, and 3.

---

> ### Author Response · Authors · 2023-11-17
> **Thank you for your time and constructive feedback**
>
> Thank you very much for the thorough feedback! We appreciate your recognition of the strengths of our paper. We address your concerns below
>
> **1. [De-emphasizing early stopping] I ask the authors to either provide more evidence on the special role of early stopping or edit the text in the abstract, introduction, and sections 5 and 6 to deemphasize the importance of early stopping.**
>
> Thank you for pointing out, we apologize for the confusion. We would first like to clarify that it is _not_ that the difference in performance between SAM and SGD is aggravated with early stopping and that is not the point of our work. Empirically, the difference is aggravated _with random label noise_ (reported in Table 4 of [1]; see our Figure 5 (Page 18) where with no label noise, SAM only observes an 8% boost in accuracy).
>
> The reason we had emphasized early stopping is the following. To characterize SAM’s label noise robustness, we want to compare the _best performance_ of SAM with the best performance of SGD. Under heavy label noise, the best test performance often occurs with early stopping for gradient based optimization methods [2]. Like SGD, SAM’s peak test accuracy also occurs before convergence as shown in Figures 1, 2, and 3.
>
> To reiterate, we aren’t claiming that early stopping has a special role in observing improvements with SAM. The point is we want to study SAM under label noise, and the best performance often occurs before convergence when mislabeled training points have not been fit.
>
> We’ve modified our submission to de-emphasize early stopping, and we hope this fix clarifies our motivation. Please let us know if the modifications are sufficient. We’re happy to make further improvements.
>
> **2. [Better depiction of faster learning of clean training examples] In addition, the presentation of the middle plot in Figure 1 is problematic: The caption says "SAM fits much more clean data however before beginning to overfit to mislabeled data" but the evidence is hard to infer from the plot. The revision should present this result more clearly.**
>
>
> We apologize for the confusion. To clarify this point, we’ve added another subfigure in Figure 1 (also in Figure 6 on Page 18) plotting the ratio of the accuracy of clean training examples over that of noisy training examples. As shown in the figure, the ratio of clean training accuracy over noisy training accuracy peaks to a noticeably higher value with SAM, and the ratio is highly correlated with the test accuracy.  Does this help present the result more clearly? We will incorporate it into the main body for the final draft.
>
> **3. [Connecting the regularization to label noise] The discussion does not properly connect this regularization effect to the overall narrative about robustness to label noise…**
>
> Thank you for the feedback! First, we reiterate the connections we do make for context. Through deriving SAM’s updates for a two layer deep linear network, we identified that SAM regularizes  the norm of the intermediate features and last layer weights (Equation 4.6). Applying a similar treatment to the last two layers in deep networks, we empirically observed that the difference in performance between SAM and SGD under label noise is more than halved, suggesting that this implicit regularization of SAM does provide robustness to label noise.
> .
>
> Admittedly, we do not provide precise theoretical analysis of the robustness effects of SAM’s network Jacobian term. Yet we find our empirical results alone quite surprising and we hope it provides interesting insights about the behavior of SAM – a large proportion of SAM’s improvements above SGD can be retrieved by a simple regularization to the last layer weights and features of the network.
>
> We contribute the following additional experimental results to strengthen the connection between the conclusion derived in 2-layer linear networks and label noise.
> - In Figure 8 (Page 19), we plot the norm of the last-layer intermediate features and weights with SAM and it is indeed noticeably smaller than SGD. Thus, a similar regularization seems to be occurring at the final two layers with SAM in deep networks.
> - Under no label noise, we find that our regularization method (with hyperparameter search) only leads to a 1% boost in accuracy (out of the 8% total performance gap between SAM and SGD) and the gain disappears at convergence. See Figure 6 (Page 18).
> \
> \
>     While under label noise, the last layer regularization closes half the SAM-SGD performance gap, the regularizer only closes an eighth of the performance gap with no label noise. This finding suggests that last-layer feature regularization, which is clearly just one aspect of SAM, is especially important under label noise.
>
> We think it would be interesting future work to make this connection theoretically precise.

---

> > ### Author Response · Authors · 2023-11-17
> > **Rebuttal Part 2**
> >
> > **3. [Unclear implications in discussion] The text below the theory only briefly says "In linear models, weight decay has somewhat equivalent effects to SAM’s logit scaling in the sense that it balances the contribution of the sample-wise gradients and thus, prevents overfitting to outliers," but I did not find any basis for this claim…**
> >
> > We apologize for the confusion. The sentence you mention was intended to discuss potential connections between SAM’s explicit up-weighting (linear analysis) to the other nonlinear effects we identified (last-layer feature/weight penalty). We will modify the paragraph (under Equation 4.5 (Page 7)) to make it clear. We expand on the connection below.
> >
> > Our analysis shows that SAM can be thought of as performing weight decay and feature regularization in the last layer. We showed that in linear models, SAM upweights the gradient of low loss points. We think this has an end implication similar to that of weight decay in linear models. In Appendix Section B, we showed that asymptotically as SAM’s rho parameter goes to infinity, SAM’s classifier converges to the empirical mean of the data scaled by the label X^T y (every training point has equal contribution). With weight decay in linear models, the solution also converges to the mean in the limit as the penalty term increases (See Equation B.9 in Appendix). This is interesting because SAM has a similar effect to weight decay but via two different mechanisms: logit upweighting in linear models and the Jacobian term’s implicit regularization in deep networks. Weight decay on its own has been shown to improve generalization error under label noise in neural networks [3].
> >
> > Please let us know if this answers your question! We are happy to make further adjustments.
> >
> > **4. The large-scale experiments in the submission are only on Cifar-10.**
> >
> > Thank you for the feedback. We’ve added additional experiments comparing SAM and SGD on flowers102 and Tiny-ImageNet with label noise reported in Figure 5 (Page 18). We can observe similar trends here where the true training accuracy of noisy examples rises to a higher value with SAM than SGD and this corresponds with better test accuracy. We will verify on more datasets to ensure SAM’s robustness to label noise is widely observable for the final paper version.
> >
> > We hope these adjustments address your main concerns. Please let us know if you have any further feedback!
> >
> > **References:**
> >
> > [1] Foret et al. (2020) Sharpness-Aware Minimization for Efficiently Improving Generalization [https://arxiv.org/abs/2010.01412]
> >
> > [2] Liu et al. (2020) Early-Learning Regularization Prevents Memorization of Noisy Labels [https://arxiv.org/abs/2007.00151]
> >
> > [3] Advani and Saxe (2017) High-dimensional dynamics of generalization error in neural networks [https://arxiv.org/abs/1710.03667]

---

> > > ### Author Response · Authors · 2023-11-21
> > > **Any further questions?**
> > >
> > > Thank you again for the comprehensive and useful review! According to your suggestions, we've
> > > 1) **De-emphasized the role of "early stopping"**: There is not direct relationship between early stopping and the gap between SAM and SGD. Rather, SAM is much better than SGD under label noise, and under label noise, the best test accuracy often occurs with early-stopping.
> > > 2) **Strengthened our nonlinear section**: Provided further results that indicate feature regularization is especially important under label noise
> > > 3) **Added results on more benchmarks**: Tiny-ImageNet and Flowers102
> > >
> > >
> > > Does this address your concerns? If you have any more questions or feedback, please let us know.

---

> > > > ### Comment · Reviewer_ATLV · 2023-11-21
> > > >
> > > > Thank you for the elaborate response. It answers my comments and, unless other issues are raised during the reviewer discussions, I will raise my score.

---

### Meta-Review · Area_Chair_DNwC · 2023-12-13

**Metareview:**

The paper investigates the generalization effect of a recently proposed smoothness-promoting optimization of overparameterized deep networks, SAM, specifically in the presence of label noise in the training data. The study is interesting since SAM has been shown to be effective in generalizing from noisy-labeled training data and its specific learning mechanisms in general is still a relatively active topic of study.


While other works have studied the normalization, batchsize, and approximation of SAM, among others, the current work studies it from a different aspect. It essentially decomposes the backward chain into two terms: (1) a loss derivative w.r.t. logits (through softmax) and then (2) the derivative of the logits w.r.t. the model parameters. It empirically shows that the second term is where the influence of SAM-like optimization becomes effective for generalization and formally argues that it can be due to the possibility of upweighting clean examples which is corroborated by empirical analysis, but shows a correlation rather than necessity.


The reviewers acknowledge the novel view and the importance of the problem but initially had some reservations, specifically with regards to the writing and lack of certain additional studies important to the statements, e.g. ruling out a similar effect in the lack of label noise, the role of early stopping, and influence of training hyperparameters that directly affect the learning dynamics.


Authors have provided a point-to-point rebuttal as well as a major revision of the paper including several new experiments. While all reviewers eventually land at a borderline rating, they are unanimously leaning towards acceptance. The AC believes that despite some notes in the observed evidences sometimes only weakly substantiating the statements, the paper has improved with the new experiments and revised writing such that the contributions are clearer. The AC does not overturn the reviewers’ unanimous decision and suggests acceptance.

**Justification For Why Not Higher Score:**

The paper's empirical or theoretical contributions are not notable enough for a spotlight paper.

**Justification For Why Not Lower Score:**

The paper has clear contributions and works on an important technique (SAM) and important task (learning in presence of label noise) which the reviewers unanimously liked. It should be accepted.

---

### Decision · Program_Chairs · 2024-01-16

Accept (poster)